# Physical Intimate Partner Violence, Childhood Physical Abuse and Mental Health of U.S. Caribbean Women: The Interrelationship of Social, Contextual, and Migratory Influences

**DOI:** 10.3390/ijerph19010150

**Published:** 2021-12-23

**Authors:** Krim K. Lacey, Regina Parnell, Sasha R. Drummond-Lewis, Maxine Wood, Karen Powell Sears

**Affiliations:** 1Department of Sociology and African and African American Studies, University of Michigan-Dearborn, Dearborn, MI 48128, USA; 2Department of Occupational Therapy, Wayne State University, Detroit, MI 48201, USA; ad9049@wayne.edu; 3Department of Behavioral Sciences, University of Michigan-Flint, Flint, MI 48502, USA; srdlewis@umich.edu; 4Department of Humanities, York University, Toronto, ON M3J 1P3, Canada; maxinew@yorku.ca; 5Department of Anthropology and Sociology, Denison University, Granville, OH 43023, USA; searsk@denison.edu

**Keywords:** childhood physical abuse, physical IPV, acculturation, discrimination, mental health

## Abstract

The literature has shown an increased risk for mental health conditions among victims of domestic violence. Few studies have examined the relationship between mental health disorders and domestic violence among Caribbean women, and how the association might be influenced by migratory and contextual factors. This study addresses the mental well-being of U.S. Caribbean Black women victims of domestic violence, and the relationships between acculturation, discrimination, and demographic influences. An analysis of data from the 2001–2003 National Survey of American Life (NSAL) re-interview, the first and most complete study on U.S. Caribbean Blacks, was conducted. Bivariate analysis revealed an association between acts of physical domestic violence and mental health conditions, with generally higher risk among women who reported both severe physical intimate partner violence and childhood physical abuse. Multivariate logistic regression indicates an association between specific mental disorders and acts of domestic violence. Acculturation, length of residence in the United States, age, education, poverty, and country of origin were also associated with mental health. The study highlights future directions for exploration including additional investigation of the influence of acculturation on the physical health of victims of domestic violence.

## 1. Introduction

The Black immigrant population has increased steadily since 1980 and is estimated to be about 9% of the Black population [1,2], with the majority being traditionally women [3]. Black women comprise 52% of the growing U.S. Black population and a high proportion of this group identifies as Sub-Saharan African or Caribbean [4]. As the number of immigrant Black women increases, the need to examine distinct cultural factors affecting their well-being becomes essential for addressing the needs of Black communities at large. Studies continue to find that Black and immigrant women are at greater risk for victimization [5,6], and the associated mental health risks are just as alarming [7]. One report estimates that half of the women in some Caribbean countries have reported violence by an intimate partner over the course of their lives [8]. Another study in the United States puts the physical intimate partner violence (IPV) rates for Black women at around 41 percent [9]. While we have made some progress in understanding more about the health consequences of victims of domestic violence, there is a paucity of research on factors that affect the well-being of U.S. Caribbean women.

In addition to intimate partner violence risks, the process of acculturation to host societies can be quite challenging and affect the well-being of immigrant women. Along with new experiences with social and structural conditions and stressors that can be consequential to mental health, some women face augmented exposure to violence (both family and intimate partner violence). The combination of the acculturation experiences and exposure to domestic abuse may place women at risk for poorer mental health outcomes. This research examines the influence of domestic violence (physical IPV and childhood physical abuse in particular) and migratory and contextual factors on the mental well-being of U.S. Caribbean Black women.

### 1.1. Background

Domestic violence across the life course in the form of adult physical intimate partner violence and childhood physical abuse is a serious and pervasive public health problem particularly within minority and immigrant communities [10,11]. Past research consistently links adult experiences with domestic violence to poor well-being [11,12,13,14]. Studies over the years have documented the association between adult intimate partner violence and a wide variety of mental health problems such as mood, anxiety, depression, and substance disorders [13,14]. Emerging yet scant research has also documented similar associations among U.S. Caribbean Black women [10]. In general, studies have also found an association between childhood abuse and mental health among adult women [15]. Documented mental disorders associated with physically victimized children include depressive disorder, suicide, PTSD, and personality disorder [16]. However, little is known about the influence of childhood abuse on the well-being of immigrant Caribbean women.

Domestic violence research shows that women of lower socioeconomic status, those living in urban areas, and those of minority status are more likely to be victimized [17]. While less studied, processes of acculturation have also been linked to domestic violence [18,19,20]. Varied studies suggest that low [21], intermediate [22,23], and even high levels of acculturation can increase the risk for intimate partner violence [24]. Given this, the likelihood for increased victimization during the acculturation process raises concerns about the potential health implications for immigrant women. A better understanding of the influence of acculturation on victimization and mental health risks becomes necessary considering the growing Black immigrant population in the United States.

Undoubtedly, migrating to a new country can be an exciting period that comes with endless possibilities to foster goal achievement and a better life [25,26]. However, for some immigrants, the process of migrating to a new country can bring a host of challenges including culture shock, family separation, and traumatic experiences with racial discrimination that might not have been experienced before migrating. These challenges can contribute to mental health problems including depression, anxiety, substance use, and poor physical conditions [27,28]. For some immigrants, poor health outcomes might be a consequence of a life of poverty in their new home as they seek to establish themselves. However, new immigrants to host countries may bring with them experiences and cultural attributes that serve as protective mechanisms and provide buffers against acculturative stress and subsequent poor health outcomes [29].

Generally, the literature suggests that the health of immigrants is initially more favorable than native-born residents early in the acculturation process and then appears to deteriorate with length of time in host countries [30,31]. The relationship between health and length of residence also holds for Caribbean immigrants to the U.S. [32,33,34]. Additionally, among immigrants to the United States gender differences exist in the health effects of acculturation with more physical symptoms, depression, smoking, and alcohol use for immigrant women than for immigrant men [28,35]. Although extant research has established the relationship between processes of acculturation and immigrant health, the interrelationship of physical domestic violence across the lifespan has rarely been examined.

### 1.2. Theoretical Frameworks

The Tridimensional Acculturation Model is useful when studying acculturative stress and well-being. The framework uses a multidimensional lens to explain the health of Caribbean Black immigrants who are seen as juggling three cultural worlds: Caribbean, African-American, and European-American [36,37]. Along with immersion within their own culture, Black immigrants are exposed to White mainstream culture as well to the culture of marginalized U.S. Black populations who face structural inequalities, including discrimination and poverty, which are known to create stress and increased risk for violence and poor health outcomes [38]. Each cultural world has a different set of challenges that require different responses, which can affect physical and emotional well-being.

Cumulative Stress Theory is also valuable when explaining the health outcomes from enduring stressors, whether due to acculturation or domestic violence. The theory posits that individuals exposed to continued adversity across their lifespan are more likely to experience diminished health and well-being than those who experience less chronic adversity [39]. The effects of stress over long periods are shown to have a cumulative negative impact on health especially among those experiencing racial discrimination and limited access to resources for coping in biased environments [40,41]. Negative health outcomes exist for all socio-economic levels but differ by gender, with Black women at an increased risk for poor health outcomes [42].

### 1.3. Research Aims

This study examined the association between mental health conditions (mood and anxiety disorders and suicide) and acts of domestic violence (i.e., severe physical IPV, childhood physical abuse) among U.S. Caribbean women. Another aim was to evaluate the combined effect of these forms of domestic violence (both forms of abuse) on the risk of particular mental health conditions. We further evaluated the interrelationship of specific acts of domestic violence, migratory factors, and contextual influences on mental health conditions. Based on extant literature, we expected to find an association between mental health disorders and specific acts of physical abuse. We also expected that the risk for mental disorders would increase with multiple victimizations. However, the nature of the interrelationships between life course domestic violence (childhood and adult), migratory factors, contextual influences, and mental health is presently unclear.

## 2. Materials and Methods

### 2.1. Participants and Sample

Data from the 2001–2003 National Survey of American Life (NSAL) re-interview were analyzed [43,44]. As part of the Collaborative Psychiatric Epidemiology Surveys (CPES), the NSAL is the most comprehensive study on the health of U.S. Blacks, and the first nationally representative study of Caribbean Blacks residing in the United States. Multistage probability sampling procedures were used to generate a sample of 6082 participants including 1623 Caribbean Blacks, a focus of this study. The Caribbean Black population was identified through two overlapping area probability sampling frames. A total of 266 Caribbean Blacks were interviewed in the NSAL core sample, while the remaining 1357 participants were selected from an area probability sample of housing units from high-density Caribbean areas. Eight primary areas were selected in five states: New York, New Jersey, Florida, Connecticut, Massachusetts, and the District of Columbia. Participants were included in the study if they were 18 years and older and were: (a) of West Indian or Caribbean descent, (b) from a Caribbean area country, or (c) had a parent or at least one grandparent who was born in a Caribbean area country [42]. Face-to-face interviews were primarily collected with a smaller proportion of interviews collected by phone. The analytic sample for the current study included 961 women of Caribbean descent with African ancestry.

### 2.2. Predictor Measures

Domestic violence. Two measures were used to address acts of domestic violence including severe physical intimate partner violence and childhood physical abuse. Severe physical intimate partner violence was assessed with the question, “Have you ever been badly beaten up by a spouse or romantic partner [34]. For childhood physical abuse, participants were asked, “As a child, were you badly beaten up by your parent or the people that raised you?” A composite of participants who reported severe physical intimate partner violence and childhood physical abuse was created and represented “both abuse” (severe physical IPV + childhood physical abuse). The measures had binary response options of “yes” and “no”.

Acculturation. We considered two indicators of acculturation appropriately aligned with U.S. and Caribbean cultures. The items were measured on a Likert scale ranging from not at all, very little, moderately, very often, and almost always. U.S. acculturation consisted of a single item where participants were asked, “How often do you identify yourselves as only American?” Caribbean acculturation consisted of eight combined items. Participants were asked, how much they do each of the following: (1) “associate with Caribbean”; (2) “enjoy listening to Caribbean music”; (3) “have contact with Caribbeans”; (4) “have Caribbean friends as a child”; (5) “family cooks Caribbean food”; (6) “have Caribbean friends”; (7) “identify themselves as only Caribbean” and (8) “identify as Caribbean American”. Altogether, the items had an internal consistency (α) of 0.89. The items were averaged for analysis.

Discrimination. Major discrimination was measured using a nine-item scale [45]. Participants were asked whether or not they had ever experienced the following events due to their race: (1) unfairly fired; (2) not been hired for a job; (3) unfairly denied a promotion; (4) unfairly stopped, searched, questioned, physically threatened or abused by police; (5) unfairly discouraged by a teacher or advising from continuing their education; (6) unfairly prevented from moving into a neighborhood because the landlord or realtor refused to sell or rent you a house or apartment (7); moved into neighborhoods where neighbors made life difficult for you and your family; (8) unfairly denied a bank loan; and (9) received service from someone such as a plumber or car mechanic that was worse than what others get. Response options of “yes” and “no” were provided. Major discrimination reflected the total count of the number of the categories of events due to race.

Control Variables. The control variables included are age (in years), poverty, education, country of origin, education, and length of residence. Poverty status is an income-to-poverty ratio measure consisting of the participants’ household income divided by the 2001 U.S. Census poverty threshold for the number of adults and children living in that household. Ratios below 1.00 indicate that the participants’ household income is below the official poverty threshold; a ratio of 1.00 or greater indicates income above the poverty level. For example, a ratio of 1.25 indicates that the income was 25 percent above the appropriate poverty threshold [46]. Education was separated into four categories: less than high school, high school graduate, some college, and college. Length of residence consists of Caribbean migrants living in the U.S. from 0 to 10 years; Caribbean migrants in the U.S. for 11 to 20 years; Caribbean migrants in the U.S. for more than 20 years; second generation consist of participants who were born in the United States to at least one Caribbean immigrant parent; and third-generation black participants who were born in the U.S. and had Caribbean born grandparents. Country of origin has four categories representing participants who had origins in English, Spanish, French, or Dutch speaking countries of the Caribbean.

### 2.3. Outcome Measures

The NSAL used a slightly modified version of a clinical assessment scale in accordance with the Diagnostic and Statistical Manual of Mental Disorders fourth edition (DSM-IV), World Health Organization Composite Interview (WHO-CIDI) to address lifetime mental disorders. In this study, we examined whether participants ever met the criteria for mood and anxiety disorders. Mood disorder was inclusive of major depressive disorder (MDD), major depressive episode (MDE), dysthymia, and bipolar disorder (any). Anxiety disorder consisted of panic, agoraphobia, generalized anxiety disorder (GAD), obsessive-compulsive disorder (OCD), and posttraumatic stress disorder (PTSD). We further evaluated suicide ideation and attempt. To address suicide ideation, participants were asked, “Have you ever seriously thought about committing suicide?” For suicide attempts, they were asked, “Have you ever attempted suicide?” The response option for both suicide measures was “yes” and “no”.

### 2.4. Analytic Strategy

Univariate analysis and bivariate tests examining the association between specific acts of domestic violence (severe physical IPV, childhood physical abuse, both) and mental health disorders were conducted. This was followed by stepwise logistic regression analysis to assess mental disorders in association with acts of domestic violence, discrimination, acculturation, and socio-demographic factors. Block 1 examined acts of domestic violence and control variables. Block 2 added acculturation and discrimination. The last block (Block 3) examined all variables. We focus on the results from the last block of the analysis. Stata 15.1 was used to conduct the analysis. The sample was weighted and corrected for standard errors, clustering, stratification, and differential non-response. An alpha of 0.05 was set for significance.

## 3. Results

### 3.1. Sample Characteristics

Caribbean women averaged 41 years of age (m = 40.5) (see Table 1). Almost three-quarters (74.4%) of women lived above the federal poverty guidelines and less than a third (30.4%) of women had at least some college education. More than seven in 10 women (71.1%) had origins in English-speaking Caribbean countries. Many participants were foreign-born (67.7%) and had been living (24.0%) in the United States between 11 and 20 years. Almost two-thirds (65.2%) of Caribbean women reside in the Northeast region of the United States. Overall, the vast majority (85.1%) of women in the sample did not experience any physical domestic violence.

### 3.2. Bivariate Analysis Examining Domestic Violence and Mental Health Disorders

Table 2 illustrates the bivariate relationship between mental health and acts of domestic violence. An association was found between acts of domestic violence and anxiety disorder (F = 18.20; *p* < 0.001); a higher rate (40%) of anxiety was found among women who reported both childhood physical abuse and adult severe physical intimate partner violence than women who experienced only one act of domestic violence or those who did not report any at all. A similar association was found for domestic violence and suicide ideation. The prevalence (53.9%) of suicide ideation was substantially higher for women who reported both severe physical IPV and childhood physical abuse than those who experience one act or no domestic violence at all (F = 10.86, *p* < 0.001). This trend continued when examining the association between domestic violence and suicide attempt (F = 16.7; *p* < 0.001). About a quarter (26.9%) of women who attempted suicide also reported experiencing both severe physical IPV and childhood physical abuse. However, the results slightly differ for mood disorders, though a significant relationship was found (F = 5.25; *p* < 0.01). A higher percentage (36.9%) of women who met the criteria for mood disorder reported only severe IPV versus 22.3 percent who reported both severe physical IPV and childhood physical abuse. The lifetime prevalence of disorders was lowest among women who reported no childhood physical abuse nor IPV.

### 3.3. Multivariate Analysis Examining Acts of Domestic Violence, Acculturation, Discrimination and Demographic Influences on Mental Health Disorders

Multivariate analysis shows that the odds (AOR = 4.51, *p* < 0.01) for mood disorder increased among participants from Spanish-speaking countries compared to English-speaking countries (see Table 3). The same was true for Caribbean women who experienced racial discrimination (AOR = 1.90, *p* < 0.05) and poverty (AOR = 2.66, *p* < 0.05). However, reduced odds (AOR = 0.211, *p* < 0.05) for mood disorders were found among Caribbean Black women who had resided in the United States between 11 and 20 years compared to those living in the country for fewer years. All things considered, acts of domestic violence were not associated with mood disorders.

The opposite was found when examining the relationship with anxiety disorders; women who reported severe physical IPV had increased odds (AOR = 4.50, *p* < 0.01) for this condition (see Table 4). The odds (AOR = 4.79, *p* < 0.001) for anxiety disorders also significantly increased among Caribbean women who lived below poverty compared to those above the federal poverty level. Similarly, participants from Spanish-speaking Caribbean countries had increased odds (AOR = 2.99, *p* < 0.05) for anxiety disorders compared to those from English-speaking countries. In the analysis, however, the odds for this disorder reduced with age (AOR = 0.962, *p* < 0.01).

The analysis examining suicide ideation showed both similarities and differences from previous models (see Table 5). The focus on suicide ideation as opposed to suicide attempts was due to sample size issues. Notably, the odds (AOR = 0.912, *p* < 0.001) for suicide ideation reduced with age. Acculturated Black women identifying with the Caribbean culture were also at reduced odds (AOR = 0.594, *p* < 0.05) for suicide ideation. Opposite to these findings, experiences with racial discrimination were associated with greater odds (AOR = 1.86, *p* < 0.01) for suicide ideation. No association between acts of domestic violence and suicide ideation was found in this model.

## 4. Discussion

This research provides further and supporting evidence of the potentially devastating health consequences that are associated with women’s exposure to domestic violence [13,14,47,48]. Bivariate analysis revealed a direct association between specific acts (severe physical IPV and childhood physical abuse) of domestic violence and mental disorders as well as suicide (both ideation and attempt). The combination of various types of physical abuse increased the risk for poorer mental health outcomes among U.S. Caribbean women, lending support to the notion that women with multiple victimizations experience the highest level of distress [49]. This was especially evident for anxiety disorders and both suicide ideation and attempt.

The relationship between physical acts of domestic violence and mental disorders by and large was supported when adjusting for demographic factors (Block 1). However, when adjusting for other factors, only specific acts of domestic violence such as severe intimate partner violence were associated with anxiety disorders. Nonetheless, other factors contributed to mental disorders. For example, experiences with poverty and discrimination were associated with the increased risk for mood and anxiety disorders, suggesting that exposure to toxic or cumulative stressors can negatively affect the mental health of U.S. Caribbean women. Women with roots in the Spanish-speaking Caribbean region also had a greater likelihood for mood and anxiety disorders. While difficult to explain, it is known that Spanish-speaking individuals have faced a unique history of stressors including racial discrimination as they acculturate to the United States [50]. In light of this finding, additional investigation is warranted. Furthermore, Caribbean Black women who had lived in the United States for 11 to 20 years were less likely to meet the criteria for mood disorders, partly supporting the healthy immigrant effect hypothesis of the initial advantage health standing that some immigrant women might maintain after their arrival to host countries.

Our study further highlights the harmful effect that poor social and stressful conditions in host countries can have on Blacks and immigrant groups; certain stressors including racial discrimination may contribute to suicide ideation. Some immigrant women may be better protected from the negative health consequences than others. For example, our findings indicated that there was a lower likelihood for suicide ideation among acculturated Caribbean women who had stronger identification with the Caribbean culture, providing some support for the tridimensional framework. This finding suggests that acculturation may serve as a buffer or protective factor against certain mental health conditions. It should be noted that there are instilled cultural beliefs in the Caribbean countries and cultures that committing suicide is a sign of weakness which might explain this finding. Finally, the risk of developing suicidal thoughts was reduced with age and could be associated with greater resilience and effective coping mechanisms that come with maturity [51,52].

### Limitations and Strength of Study

We recognize that this study is not without limitations and would like to highlight a few for consideration when interpreting the findings. Among the limitations of note is the age of the data. The data were collected during a time where social and political conditions may have differed from today. For example, we speculate that certain social movements (i.e., #MeToo) may influence victimized women’s disclosure of violence and possible outcomes. Although the dataset is more than a decade old, to our knowledge the NSAL remains the most comprehensive representative sample on the U.S. Caribbean population that allows for addressing the research objectives. Additionally, the study focused only on physical violence. Other types of violence (i.e., psychological) were not included in the sample because of the manner in which the data were collected. It is also important to keep in mind that questions surrounding domestic violence (i.e., child abuse) are retrospective and are subject to recall bias which might affect the validity of the findings. Finally, as with cross-sectional samples, causal inferences cannot be drawn.

Regardless of the limitations discussed, this study contributes to the literature in many ways. To begin, it is one of few studies based on representative data that examines the interrelationship of domestic violence, migratory factors, and contextual influences on the mental health of Caribbeans residing within the United States. The study further examined mental disorders using a clinical assessment scale which is limited in domestic violence studies on the Caribbean population. Importantly, this study is one of few to examine the mental health of Caribbean women who reported childhood physical abuse. Finally, our study utilized valid measures of acculturation which have been limited in studies of this nature. Many studies are based on proxies (i.e., language), which may not provide a valid assessment of acculturation

## 5. Conclusions

Our research shows that domestic violence can adversely affect the health and well-being of Caribbean women, whether in the form of physical intimate partner violence or childhood physical abuse. This research further sheds light on the influence that migratory factors and poor social and structural conditions can have on women’s well-being which adds to our understanding in finding effective prevention and intervention strategies. Irrespective of the rather innovative approach in utilizing improved measures of acculturation, it is also vital that we examine other measures of acculturation in future studies. Furthermore, these measures should be considered in research that examines the physical health disposition of immigrant victims of violence. These studies are particularly necessary as research suggests an increase in victimization with different levels of acculturation. Independent of these important steps, our study has highlighted subsequent findings that warrant further exploration including the effects of multiple types of physical violence on the well-being of Caribbean women.

## Figures and Tables

**Table 1 ijerph-19-00150-t001:** Sample Characteristics (N = 961).

Characteristics	Std. Err	Percentage
**Mean age (mean)**	1.176	40.5
**Poverty**		
Below-at	0.0257	25.6
Above	0.0257	74.4
**Education Status**		
Less than HS	0.0197	19.4
HS Graduate	0.0247	29.5
Some College	0.0299	30.4
College	0.0143	20.8
**Region**		
Northeast	0.0540	65.2
Midwest	0.0222	3.9
South	0.0600	25.0
West	0.0280	5.9
**Country of Origin (Language) ^1^**		
Spanish	0.0217	13.7
French	0.0250	14.3
English	0.0367	71.1
Dutch	0.0054	0.96
**Length of Residence**		
0–10 years	0.1518	15.2
11–20 years	0.2401	24.0
>20 years	0.2851	28.5
Second Generation	0.0297	21.7
Third Generation	0.0220	10.6
**Immigrant Status**		
US Born	0.0286	32.3
Foreign-Born	0.0286	67.7
**Abuse**		
No Abuse	0.0216	85.1
Childhood Abuse Only	0.0048	2.9
IPV Only	0.0178	9.3
Both IPV and Childhood Abuse	0.0105	2.7

^1^ Note. **English-speaking countries:** Jamaica, Barbados, Guyana, Trinidad and Tobago, Anguilla, Antigua, Bahama, Bermuda, British Virgin Islands, Tortola, Cayman Islands, Dominica, Grenada/Grenadines, Montserrat, St Kitts-Nevis, St. Lucia, St. Vincent, Turks and Caicos, US Virgin Island, St. Croix, St. Thomas, West. Indies, and British W. Indies. **Spanish-speaking countries:** Puerto Rico, Dominican Republic, Cuba, Panama, Costa Rica, and Nicaragua. **French-speaking countries:** Haiti, French Guiana, Guadalupe, and Martinique. **Dutch-speaking countries**: Aruba, St. Eustatius, St. Maarten, Suriname, and Netherland Antilles.

**Table 2 ijerph-19-00150-t002:** Bivariate Analysis of Domestic Violence and Mental Disorders and Behaviors Among Caribbean Black Women.

Domestic Violence	Anxiety Disorder		Mood Disorder		Suicide Ideation		Suicide Attempt	
	No	Yes	No	Yes	No	Yes	No	Yes
**No Abuse**	88.7	11.3	85.8	14.2	91.3	8.8	98.8	1.2
**Child Abuse (Only)**	79.4	20.6	75.4	24.6	88.6	11.5	93.8	6.2
**IPV (Only)**	63.3	36.7	63.1	36.9	79.1	20.9	90.8	9.3
**Both Child Abuse and IPV**	60.1	40.0	77.7	22.3	46.1	53.9	73.2	26.9
**F-test**		18.20		5.25		10.86		16.7
***p*-value**	***	0.0000	**	0.0086	***	0.0001	***	0.0000

** *p* < 0.01; *** *p* < 0.001.

**Table 3 ijerph-19-00150-t003:** Multivariate Analysis Examining Domestic Violence, Acculturation, Discrimination and Demographic Factors on Lifetime Mood Disorder.

Variables	Block 1	Block 2	Block 3
**Age**	0.961 (0.936–0.984) **	0.972 (0.936–101)	0.970 (0.932–1.01)
**Place of origin (language)**			
English	1	1	1
Spanish	3.25 (1.62–6.50) **	3.90 (1.26–12.09) *	4.51 (1.49–13.67) **
French	0.801 (0.277–2.32)	1.56 (0.408–5.96)	1.71 (0.469–6.24)
Dutch	0.246 (0.014–4.33)	0.144 (0.003–5.63)	0.141 (0.004–4.73)
**Length of residence**			
0–10 years	1	1	1
11–20 years	1.12 (0.454–2.75)	0.207 (0.058–0.753) *	0.211 (0.061–0.753) *
>20 years	2.70 (1.09–6.68) *	1.46 (0.368–5.80)	1.17 (0.233–5.90)
Second Gen	1.88 (0.619–5.70)	0.759 (0.198–2.91)	0.766 (0.184–3.18)
Third Gen	8.57 (2.43–30.27) **	3.71 (0.509–27.03)	2.57 (0.331–19.99)
**Education**			
Less than HS	1	1	1
HS Graduate	1.08 (0.321–3.64)	1.24 (0.166–9.30)	0.745 (0.100–5.53)
Some College	2.65 (1.04–6.71) *	2.31 (0.485–11.04)	1.56 (0.307–7.89)
College	3.32 (0.836–13.21)	3.40 (0.326–35.37)	2.30 (0.192–27.53)
**Poverty**			
Above Poverty	1	1	1
Below	1.19 (0.687–2.07)	3.65 (1.69–7.88) **	2.66 (1.12–6.30) *
**Discrimination**		1.70 (1.12–2.58) *	1.90 (1.11–3.26) *
**Caribbean Acculturation**		0.958 (0.552–1.66)	0.953 (0.549–1.65)
**U.S. Acculturation**		0.911 (0.647–1.28)	891 (0.622–1.28)
**IPV**			
No	1		1
Yes	3.49 (1.41–8.65) **		2.12 (0.351–12.80)
**Childhood Abuse**			
No	1		1
Yes	2.37 (0.866–6.47) *		1.08 (0.126–9.22)
**Both Abuse**			
No	1		1
Yes	0.201 (0.047–0.849) *		0.053 (0.002–1.42)

*p* < 0.10; * *p* < 0.05; ** *p* < 0.01.

**Table 4 ijerph-19-00150-t004:** Multivariate Analysis Examining Impact of Domestic Violence, Acculturation, Discrimination and Demographic Factors on Lifetime Anxiety Disorders.

Variables	Block 1	Block 2	Block 3
**Age**	0.967 (0.942–0.992) **	0.969 (0.945–0.993) **	0.962 (0.935–0.990) **
**Place of origin (language)**			
English	1	1	1
Spanish	1.69 (0.946–3.00)	3.04 (1.21–7.64) *	2.99 (1.11–8.10) *
French	0.807 (0.370–1.76)	0.467 (0.086–2.55)	0.525 (0.106–2.59)
Dutch	---	---	---
**Length of residence**			
0–10 years	1	1	1
11–20 years	0.720 (0.351–1.48)	0.758 (0.191–3.00)	0.710 (0.185–2.72)
>20 years	1.88 (0.767–4.63)	2.10 (0.495–8.97)	1.64 (0.398–6.74)
Second Gen	1.20 (0.525–2.73)	1.82 (0.466–7.11)	1.62 (0.490–5.37)
Third Gen	2.62 (0.950–7.24)	1.94 (0.371–10.10)	1.13 (0.219–5.82)
**Education**			
Less than HS	1	1	1
HS Graduate	1.21 (0.495–2.94)	3.17 (0.820–12.23)	2.78 (0.569–13.56)
Some College	1.54 (0.637–3.71)	1.56 (0.349–6.93)	1.60 (0.298–8.61)
College	1.17 (0.573–2.35)	1.96 (0.667–5.72)	2.30 (0.564–9.40)
**Poverty**			
Above Poverty	1	1	1
Below	2.42 (1.27–4.61) **	6.19 (2.42–15.82) ***	4.79 (1.96–11.70) ***
**Discrimination**		1.39 (0.960–2.01)	1.28 (0.869–1.89)
**Caribbean Acculturation**		1.02 (0.614–1.71)	0.977 (0.580–1.64)
**U.S. Acculturation**		0.985 (0.776–1.25)	0.969 (0.746–1.26)
**IPV**			
No	1		1
Yes	4.23 (2.43–7.35) ***		4.50 (1.44–14.06) **
**Childhood Abuse**			
No	1	1	1
Yes	2.75 (1.03–7.38) *		2.55 (0.400–16.20)
**Both Abuse**			
No	1	1	1
Yes	0.528 (0.139–2.01)		0.117 (0.012–1.13)

** p* < 0.05; ** *p* < 0.01; *** *p* < 0.001.

**Table 5 ijerph-19-00150-t005:** Multivariate Analysis Examining Domestic Violence, Acculturation, Discrimination and Demographic Factors on Lifetime Suicide Ideation.

Variables	Block 1	Block 2	Block 3
**Age**	0.937 (0.909–0.966) ***	0.910 (0.875–0.947) ***	0.912 (0.877–.948) ***
**Place of origin (language)**			
English	1	1	1
Spanish	1.41 (0.483–4.10)	1.70 (0.386–7.54)	1.24 (0.247–6.24)
French	0.993 (0.266–3.71)	1.81 (0.312–10.51)	2.03 (0.406–10.20)
Dutch	---	---	---
**Length of time/Gen**			
0–10 years	1	1	1
11–20 years	1.31 (0.583–2.95)	0.923 (0.368–2.31)	0.713 (0.276–1.84)
>20 years	3.29 (1.13–9.55) *	4.29 (0.951–19.36)	4.20 (0.916–19.22)
Second Gen	2.87 (1.18–6.93) *	1.18 (0.310–4.48)	1.17 (0.207–6.68)
Third Gen	7.05 (1.70–29.22) **	1.05 (0.310–4.48)	1.20 (0.098–14.66)
**Education**			
Less than HS	1	1	1
HS Graduate	1.39 (0.538–3.59)	1.01 (0.174–5.81)	1.17 (0.163–8.33)
Some College	0.573 (0.228–1.44)	0.080 (0.011–0.589) *	0.133 (0.017–1.05)
College	1.48 (0.522–4.19)	1.00 (0.159–6.27)	1.72 (0.254–11.61)
**Poverty**			
Above Poverty	1	1	1
Below	0.532 (0.247–1.15)	0.602 (0.183–1.98)	0.534 (0.165–1.72)
**Discrimination**		2.38 (1.54–3.68) ***	1.86 (1.15–3.00) **
**Caribbean Acculturation**		0.571 (0.324–1.01)	0.594 (0.356–0.991) *
**U.S. Acculturation**		1.20 (0.788–1.84)	1.19 (0.760–1.86)
**IPV**			
No	1	1	1
Yes	2.23 (0.916–5.45)		3.84 (0.663–22.22)
**Childhood Abuse**			
No	1		1
Yes	1.41 (0.330–6.02)		3.07 (0.254–37.23)
**Both Abuse**			
No	1		1
Yes	4.22 (0.652–27.39)		0.982 (0.041–23.62)

** p* < 0.05; ** *p* < 0.01; *** *p* < 0.001.

## Data Availability

The public use NSAL data is archived at the University of Michigan-ICPSR.

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
