# Peer review of "Physical Intimate Partner Violence, Childhood Physical Abuse and Mental Health of U.S. Caribbean Women: The Interrelationship of Social, Contextual, and Migratory Influences"

_ijerph, 2021, doi:10.3390/ijerph19010150_

Round 1

Reviewer 1 Report

This article makes an important contribution to the literature. It addresses childhood sexual abuse, intimate partner violence, mental health problems, acculturation issues, and their impact on U.S. Caribbean Black women. It is critical to add the experience of Black women to what is known about subpopulations of victims/survivors of abuse.

When I read the article I at first wondered why the DSM-4 was used instead of the DSM-5. It appears only in the Limitations and Strengths of Study section regarding the age of the data. Actually I had to look back to Section 2 on Materials and Methods to the first sentence. That is where references 43 and 44 reveal earlier publications from the data in 2004 and 2007. The authors need to be more upfront about the age of the data and be clear why this is still relevant. Can the authors discuss what changes there might be in the years since the National Survey of American Life? This is a significant weakness of the article but does not have to be cause for rejection.

In the paragraph on acculturation on page 4, the question is asked "How often do you identify themselves as only American?" Should it be "yourselves"?

Author Response

Reviewer #

Comments and Response

1

When I read the article I at first wondered why the DSM-4 was used instead of the DSM-5. It appears only in the Limitations and Strengths of Study section regarding the age of the data. Actually I had to look back to Section 2 on Materials and Methods to the first sentence. That is where references 43 and 44 reveal earlier publications from the data in 2004 and 2007. The authors need to be more upfront about the age of the data and be clear why this is still relevant.

We thank the reviewer for their suggestion regarding the transparency of the age of the data. We have since included the year of data collection for the sample in the abstract and methods section of the paper.

1

Can the authors discuss what changes there might be in the years since the National Survey of American Life? This is a significant weakness of the article but does not have to be cause for rejection.

It is unclear what specifically the reviewer is referring to, whether this pertains to the DSM measures used or the social movements that have occurred since data collection. We acknowledge as suggested that we have moved from the DSM-4 to currently DSM-5 which may have an affect on diagnosis. Importantly, there has been certain movements (i.e., #metoo) that may have an impact of rate of disclosure of violence and subsequent outcomes.

1

In the paragraph on acculturation on page 4, the question is asked "How often do you identify themselves as only American?" Should it be "yourselves"?

We thank the reviewer for the editorial suggestion. The change has been made within the manuscript to reflect “yourselves”

Reviewer 2 Report

The interesting and novel part of this paper is the focus on acculturation, discrimination and demographic influences regarding victims of domestic violence. However, only two measures were used to address acts of domestic violence: severe physical intimate partner violence and childhood physical abuse. Physical violence is only a part of domestic violence and this focus of the study should be clear in the title of the paper, in the introduction, in the results and in the discussion and conclusions. Even the keywords need to be rewritten to make this clear to the reader. This omission results in a low quality of presentation and low scientific soundness. There are many studies that have been conducted on the consequences of physical violence in childhood and adulthood and these are not reported. This must be improved and the title must be changed because the present title gives the wrong impression of what the study is about (an idea would be Consequences of Physical Violence in Childhood and Adulthood for U.S. Caribbean Women: ....). The whole paper must be rewritten with this focus on physical violence in mind.

Another weakness of the paper is that the authors use domestic violence, domestic abuse, life course abuse and intimate partner violence, interchangeably without a proper definition of these terms. This should be easy to do.

The authors themselves identify another limitation of the study and that is the age of the data because the dataset is more than a decade old. Because of the #MeToo Movement there has been a major change in the discourse about violence against women and if the data were gathered in this or in the very recent years the results would have been very different I am sure because women are more ready to disclose violence against them than ever before, even physical violence. I don't know if the authors are saving other types of violence in the data for a later paper. At least make it clear in the beginning that you are focusing on physical violence. No shame in that just make it clear. However, you need to report the research results of the many research papers focusing on physical violence, both in childhood and in adulthood.

An interesting finding is that acculturated Black women identifying with the Caribbean culture were at reduced odds for suicide. 

Author Response

Reviewer #

Comments and Response

2

The interesting and novel part of this paper is the focus on acculturation, discrimination and demographic influences regarding victims of domestic violence. However, only two measures were used to address acts of domestic violence: severe physical intimate partner violence and childhood physical abuse. Physical violence is only a part of domestic violence and this focus of the study should be clear in the title of the paper, in the introduction, in the results and in the discussion and conclusions. Even the keywords need to be rewritten to make this clear to the reader. This omission results in a low quality of presentation and low scientific soundness. There are many studies that have been conducted on the consequences of physical violence in childhood and adulthood and these are not reported. This must be improved and the title must be changed because the present title gives the wrong impression of what the study is about (an idea would be Consequences of Physical Violence in Childhood and Adulthood for U.S. Caribbean Women: ....). The whole paper must be rewritten with this focus on physical violence in mind.

We appreciate the reviewer’s perspective and have attempted to address their suggestions and concerns. As such, we have adjusted the title to reflect the measures used to address domestic violence in this study. In addition, we have added literature geared to the consequences of and childhood violence. This is reflected in the background section and the reference section of the manuscript.

2

Another weakness of the paper is that the authors use domestic violence, domestic abuse, life course abuse, and intimate partner violence, interchangeably without a proper definition of these terms. This should be easy to do

We have changed the wording to maintain consistency throughout the paper. We’ve opted to focus on domestic violence which is inclusive of intimate partner violence and childhood abuse. This is noted in the first line of the background section of the manuscript. Whenever necessary, we report the specific act of violence in relation to the mental health of Caribbean Black women.

2

The authors themselves identify another limitation of the study and that is the age of the data because the dataset is more than a decade old. Because of the #MeToo Movement, there has been a major change in the discourse about violence against women and if the data were gathered in this or in the very recent years the results would have been very different I am sure because women are more ready to disclose violence against them than ever before, even physical violence. I don't know if the authors are saving other types of violence in the data for a later paper. At least make it clear in the beginning that you are focusing on physical violence. No shame in that just make it clear. However, you need to report the research results of the many research papers focusing on physical violence, both in childhood and in adulthood.

We agree with the reviewer and appreciate their comment to some extent. As such, we have since made an effort to make the readers aware that the focus of the paper is on physical intimate partner violence and physical childhood abuse by making this known throughout the paper including the title, introduction, result, discussion, and conclusion section of the manuscript. Supporting literature has also been added that focuses on physical intimate partner violence and childhood abuse.

Secondly, even though we tend to agree that things have changed socially and politically (e.g., #metoo) since the data collection that could affect victimized women’s response and outcome to intimate partner violence, it is unknown to what extent. It should be kept in mind that there are communities and cultural groups where intimate partner remains a private matter and therefore might not readily conform to these changes. This is reflected by the increase in violence since the #metoo movement and the few changes made to address this issue among certain populations (i.e., Caribbean) (Levy & Mattsson, 2021; Hallet, 2019). Furthermore, it has been noted that within certain administrations, women (undocumented) were reluctant to come forward due to fear of reprisals including deportation. Even so, we have noted in the manuscript that it is possible that recent social and political changes since the time of data collection could influence victimized women's responses and outcomes.

Finally, the measure used to address physical intimate partner violence and childhood abuse are the only measures within the dataset representing domestic violence. Data on other types of domestic violence were not collected or included in the sample.  

2

An interesting finding is that acculturated Black women identifying with the Caribbean culture were at reduced odds for suicide.

We thank the reviewer for their comment. We agree that this is an interesting finding as well and have provided possible explanations surrounding it.

Reviewer 3 Report

The paper “Domestic Violence and the Mental Health of U.S. Caribbean Women: The Interrelationship of Social, Contextual, and Migratory Influences” examines the associations between mental health and experienced domestic violence among US Caribbean women. In addition, combined effects of specific types of domestic violence (childhood abuse, intimate partner violence) on the risk of particular mental health conditions are examined. Interrelationships of domestic violence, migratory factors and contextual influences on mental health are also explored. The main strengths and contributions of this article are: 1) representative sample for the exploration of this topic, 2) usage of clinical assessment scales for mental disorders in a study that explores domestic violence on Caribbean samples, 3) data on mental health of Caribbean women who experienced childhood abuse, 4) usage of valid measure of acculturation. I think that the topic of the paper is very interesting, important and valuable to be published in this journal.

The paper has an appropriate structure of the original empirical research paper and contains abstract with keywords, introduction, materials and methods, results, discussion, conclusions and references. The abstract is well written and contains relevant information, and the keywords represent the topic that is in focus of the paper. Introduction has relevant review of empirical findings related to the topic of the paper, and theoretical frameworks (Tridimensional Acculturation Model, and Cumulative Stress Theory) for the research are also well explained. Introductory part also contains research aims that are in line with abstract, methodology as well as results and discussion. The gap in knowledge is explicitly stated, and the references used are appropriate.

Methodology part of the text contains the information on participants and sampling. I would advise to add the subtitle “Participants”, “Sample” or “Participants and sampling” for the first part of the methodology section. Measures, both predictor variables as well as outcome variables, are well described in the following subsections. Methods sections ends with the Analytic strategy, where all analyses conducted for the purpose of this paper have been described.

Empirical results are well described, and this section consists of 5 tables. Although the key results are well described, I have two comments on this part. First, I would appreciate if it would be explicitly stated that only results for the Block 3 are interpreted for the Tables 3 to 5 (and not Blocks 1 and 2). And also, I have noticed an error in the interpretation of the Table 4 related to the effect of the type of domestic violence on anxiety disorder. The text states that “women who reported child abuse had increased odds (AOR=4.50, p < .01) for this condition”, but the Table 4 shows, as I understand, that IPV has significant effect on anxiety disorder (AOR=4.50, p < .01). This same mistake can also be seen in discussion, and should be corrected. Instead of “only specific acts of domestic violence such as child abuse were associated with anxiety disorder”, should be stated “only specific acts of domestic violence such as intimate partner violence were associated with anxiety disorder”.

Discussion, which contains limitation and strengths of the study, as well as conclusion part, are well written, and add to the quality of this valuable paper. Limitations or weaknesses of the study are explicitly stated, and include: 1) dataset older than 10 years, 2) measure of violence only includes physical violence and not other forms of violence, 3) data are retrospective (e.g. on child abuse) which could affect validity of findings, 4) study design is cross-sectional which does not allow causal inferences.

The list of references contains 51 reference, and 30 of them are recent and published within the last 10 years, and overall 13 references are published within the last 5 years. Information about the year of publication is missing in two references and should be added (reference No 32 and No 36).  There are only 3 self-citations.

I recommend acceptation of the paper with minor changes.

These are my specific comments and suggestions for the improvement of the manuscript:

Page 7 Please explicitly state that only results for the Block 3 are interpreted for the Tables 3 to 5.

Page 8 Correct the text that describes the Table 4. Instead “women who reported child abuse had increased odds (AOR=4.50, p < .01) for this condition”, the text should state “women who reported IPV had increased odds (AOR=4.50, p < .01) for this condition”. (if the results in Table 4 are correct)

Page 10 Instead of “only specific acts of domestic violence such as child abuse were associated with anxiety disorder”, should be stated “only specific acts of domestic violence such as intimate partner violence were associated with anxiety disorder”. (if the results in Table 4 are correct)

Page 13 Information about the year of publication is missing in two references and should be added (reference No 32 and No 36).

Author Response

Reviewer #

Comments and Response

3

Methodology part of the text contains the information on participants and sampling. I would advise to add the subtitle “Participants”, “Sample” or “Participants and sampling” for the first part of the methodology section. Measures, both predictor variables as well as outcome variables, are well described in the following subsections. Methods sections ends with the Analytic strategy, where all analyses conducted for the purpose of this paper have been described.

We thank the reviewer for their suggestion. The subtitle has been added to reflect “Participants and Sample”

3

Page 7 Please explicitly state that only results for the Block 3 are interpreted for the Tables 3 to 5.

We thank the reviewer for their helpful comment. It is now noted in the analytic strategy section of the paper that the final block of the analysis is reported/interpreted for tables 3 to 5.

3

Page 8 Correct the text that describes the Table 4. Instead “women who reported child abuse had increased odds (AOR=4.50, p < .01) for this condition”, the text should state “women who reported IPV had increased odds (AOR=4.50, p < .01) for this condition”. (if the results in Table 4 are correct)

This has now been corrected to reflect the reviewer’s suggestion. We appreciate the reviewer drawing our attention to this important oversight.

3

Instead of “only specific acts of domestic violence such as child abuse were associated with anxiety disorder”, should be stated “only specific acts of domestic violence such as intimate partner violence were associated with anxiety disorder”.

The reviewer’s suggestion is now included. We thank the reviewer for their suggestion

3

Discussion, which contains limitation and strengths of the study, as well as conclusion part, are well written, and add to the quality of this valuable paper. Limitations or weaknesses of the study are explicitly stated, and include: 1) dataset older than 10 years, 2) measure of violence only includes physical violence and not other forms of violence, 3) data are retrospective (e.g. on child abuse) which could affect validity of findings, 4) study design is cross-sectional which does not allow causal inferences.

We thank the reviewer for their thoughtful comments

3

Page 13 Information about the year of publication is missing in two references and should be added (reference No 32 and No 36).

The changes have been made to the references outlined by the reviewer. We thank the reviewer for their helpful comments

Round 2

Reviewer 2 Report

I suggest a slight change in the title:

Physical Intimate Partner Violence, Childhood Physical Abuse and Mental Health of U.S. Caribbean Women: The Interrelationship of Social, Contextual, and Migratory Influences

Abstract: line 8. acts of physical domestic violence

Page 2, line 8: childhood physical abuse

Page 2. 1.1. Background: line 2: and childhood physical abuse

Page 2. 1.2. Theoretical Frameworks : move the title to next page

Author Response

Reviewer #

Comments and Response

2

I suggest a slight change in the title: Physical Intimate Partner Violence, Childhood Physical Abuse and Mental Health of U.S. Caribbean Women: The Interrelationship of Social, Contextual, and Migratory Influences

The adjustment to the title has been changed to reflect the reviewer’s suggestion

2

I suggest a slight change: Abstract: line 8. acts of physical domestic violence

A slight change to the abstract has been made with the word “physical” added

2

I suggest a slight change: Page 2, line 8: childhood physical abuse

The word “physical” has been added Page 2, line 8

2

I suggest a slight change: Page 2. 1.1. Background: line 2: and childhood physical abuse

The word “physical” has been added in line 2

2

I suggest a slight change: Page 2. 1.2. Theoretical Frameworks: move the title to next page

The title “Theoretical Framework” has been moved to the next page